# Active and pH-Sensitive Nanopackaging Based on Polymeric Anthocyanin/Natural or Organo-Modified Montmorillonite Blends: Characterization and Assessment of Cytotoxicity

**DOI:** 10.3390/polym14224881

**Published:** 2022-11-12

**Authors:** Tomy J. Gutiérrez, Ignacio E. León, Alejandra G. Ponce, Vera A. Alvarez

**Affiliations:** 1Grupo de Materiales Compuestos Termoplásticos (CoMP), Instituto de Investigaciones en Ciencia y Tecnología de Materiales (INTEMA), Facultad de Ingeniería, Universidad Nacional de Mar del Plata (UNMdP) y Consejo Nacional de Investigaciones Científicas y Técnicas (CONICET), Colón 10850, Mar del Plata B7608FLC, Argentina; 2Centro de Química Inorgánica “Dr. Pedro J. Aymonino” (CEQUINOR), Facultad de Ciencias Exactas, Universidad Nacional de La Plata (UNLP) y Consejo Nacional de Investigaciones Científicas y Técnicas (CONICET), Blvd. 120 No. 1465, La Plata 1900, Argentina; 3Grupo de Investigación en Ingeniería en Alimentos (GIIA), Instituto de Ciencia y Tecnología de Alimentos y Ambiente (INCITAA, CIC-UNMDP), Facultad de Ingeniería, Universidad Nacional de Mar del Plata, Juan B. Justo 4302, Mar del Plata B7602AYL, Argentina

**Keywords:** active compounds, food additives, food toxicity, functional foods, nano-encapsulation

## Abstract

Polymeric anthocyanins are biologically active, pH-sensitive natural compounds and pigments with beneficial functional, pharmacological and therapeutic properties for consumer health. More recently, they have been used for the manufacture of active and pH-sensitive (“intelligent”) food nanopackaging, due to their bathochromic effect. Nevertheless, in order for polymeric anthocyanins to be included either as a functional food or as a pharmacological additive (medicinal food), they inevitably need to be stabilized, as they are highly susceptible to environmental conditions. In this regard, nanopackaging has become a tool to overcome the limitations of polymeric anthocyanins. The objective of this study was to evaluate their structural, thermal, morphological, physicochemical, antioxidant and antimicrobial properties, as well as their responses to pH changes, and the cytotoxicity of blends made from polymeric anthocyanins extracted from Jamaica flowers (*Hibiscus sabdariffa*) and natural or organo-modified montmorillonite (Mt), as active and pH-sensitive nanopackaging. This study allowed us to conclude that organo-modified Mts are efficient pH-sensitive and antioxidant nanopackaging systems that contain and stabilize polymeric anthocyanins compared to natural Mt nanopackaging and stabilizing polymeric anthocyanins. However, the use of these polymeric anthocyanin-stabilizing organo-modified Mt-based nanopackaging systems are limited for food applications by their toxicity.

## 1. Introduction

Polymeric anthocyanins are biologically active and pH-sensitive natural pigments that have attracted the attention of food scientists and technologists globally, as they have important beneficial effects on consumer health [1,2]. In particular, polymeric anthocyanins extracted from Jamaica flowers (*Hibiscus sabdariffa*) have been shown to reduce cholesterol levels, increase lipid peroxidation inhibition capacity [3], prevent kidney diseases [4] and reduce hypertension in patients with type 2 diabetes [5]; they have antimicrobial [6] and antioxidant [3] properties, and crucially, they are non-toxic [3]. These characteristics make Jamaica flower extract (JFE) a promising food additive for the development of functional and/or medicinal foods. In addition, polymeric anthocyanins, in general, have contributed to the manufacture of pH-sensitive composite food packaging [7,8,9,10]. These pH-sensitive packaging materials that contain polymeric anthocyanins have been normally classified as “intelligent packaging”, since they can act as sensors that are capable of providing information on the status of food quality and safety [11]. For example, Merz et al. [12] utilized anthocyanins that were extracted from jambolan (*Syzygium cumini*) fruit to develop pH-sensitive packaging materials (colorimetric indicator/chromatic sensor) for the purpose of monitoring shrimp quality and safety. This fact was justified, since biogenic amines (e.g., histamine–allergenic compounds) that are generated as secondary metabolites from histidine (essential amino acid) during the decomposition of fishery products can modify the structure and coloration of polymeric anthocyanins, thus providing early warnings for food quality and safety [13,14,15,16,17,18]. Notably, Jamaica flowers are traditionally marketed to make polymeric anthocyanin-rich infusions or teas, and the main polymeric anthocyanin contained in JFE is delphinidin-3-O-sambubioside [3].

However, in order for polymeric anthocyanins to be useful to the food industry, they must be stabilized, as they are susceptible to light, oxygen, temperature and the presence of enzymes [19,20]. One way of doing this is by nanopackaging the active compound using ionic gelation, with pectin as a wall biopolymer [21]. In fact, the nanopackaging of various active compounds with different biopolymers has frequently been employed, in order to develop safe functional foods [22]. Notwithstanding, the stabilization and nanopackaging of polymeric anthocyanins extracted from blueberries [23] and acerola [20] has been carried out more recently through the use of nano-clays, mainly montmorillonite (Mt). Nonetheless, these bio-nanocomposite systems have not yet been evaluated in terms of their toxicity.

According to the literature, up until now there has been only one study that has evaluated the toxicity of Mt-based nanopackaging composite materials as food additives [24], and we feel that more research is required to evaluate their true potential as polymeric anthocyanin nanopackaging materials for the food industry.

Polymeric anthocyanins can be stabilized through the use of Mt by intercalating the active compound between the aluminosilicate layers. In this way, bio-nanocomposite systems are generated that protect the polymeric anthocyanins from different processing conditions [20,23].

The two main aims of this study were as follows: (1) to evaluate and characterize different systems of chemically modified Mts, regarding their ability to stabilize and nanopackage polymeric anthocyanins extracted from Jamaica flowers; and (2) to assess possible toxicological effects in order to determine whether these bio-nanocomposite systems can be applied as nanopackaging materials in the food industry.

## 2. Experimental

### 2.1. Materials

Polymeric anthocyanins (antioxidant and pH-sensitive pigment) obtained from Jamaica flower (*Hibiscus sabdariffa*) extract (JFE) were added to three different montmorillonites (Mts), with the aim of developing potentially active and pH-sensitive nanopackaging. All of the Mts tested were supplied by Laviosa Chimica Mineraria S.p.A. (Livorno, Italy): natural Mt (NMt, or HPS according to the manufacturer), Mt modified with dimethyl benzylhydrogenated tallow ammonium (MtMB, denominated 43B by the manufacturer) and Mt modified with dimethyl dihydrogenated tallow ammonium (MtMD, or 72T according to the manufacturer). Following the manufacturer’s descriptions, these modified Mts were nano-clays that were prepared from a naturally occurring Mt, purified and modified with quaternary ammonium salts (dimethyl benzylhydrogenated tallow ammonium and dimethyl dihydrogenated tallow ammonium). All Mts were used as received. The cation exchange capacity (CEC) of the Mts, as measured by the methylene blue method [25], yielded a CEC of 128 meq/100 g clay. JFE (100% polymeric anthocyanin—average molecular weight ranging from 600 to 10,000 Da) was obtained according to the methodology proposed by Dai et al. [26] using ethanol as a solvent, since it maintains the properties of polymeric anthocyanin-rich extracts. Briefly, dehydrated Jamaica flowers were purchased from a local market in the Ciudad Autónoma de Buenos Aires, Provincia de Buenos Aires, Argentina. The flowers were marketed by an Argentinian company, and were labelled as harvested in the Jardín America (Street Amado Nervo 478, Provincial de Misiones, Argentina; geographical coordinates—latitude S 27°2′25.597″ and longitude W 55°14′22.906″). A total of 30 g of Jamaica flowers were weighed and immediately immersed in 200 mL of ethanol (Aldrich, St. Louis, MO, USA—product code: 34923), applying a slight pressure on the immersed flowers in order to extract the polymeric anthocyanins. The extract was then decanted to remove the solid fragments of the flower petals. The JFE and the JFE-containing Mts were developed on the same day, and kept refrigerated at 5 °C in a dark container until further processing, in order to avoid oxidative damage.

### 2.2. Manufacture of Potential Active and pH-Sensitive Nanopackaging

JFE was added to the Mts by manually blending 20 g of each Mt with 50 mL of JFE at room temperature (21 °C). The blend was then frozen at −20 °C for 48 h, after which it was lyophilized at 13.33 Pa (100 mTorr) and then frozen again at −50 °C for 72 h, utilizing a Gland type Vacuum Freeze Dryer, Columbia International, Model FD-1B-50 (Shaan Xi, China), in order to obtain a solvent-free product. Lyophilization also preserves the polymeric anthocyanins (active compound) in the JFE, and ensures JFE-containing nano-sized Mt particles are obtained. The resulting JFE-containing Mts were conditioned in containers with a saturated solution of NaBr (*a_w_*~0.575 at 25 °C) for seven days before testing, in order to maintain controlled and known conditions. During this period, the containers were protected from light in a dark room, in order to avoid the photodegradation of the polymeric anthocyanins. Six Mt nanosystems were manufactured and labeled as follows: natural Mt (NMt), natural Mt-containing JFE (NMt + JFE), Mt modified with dimethyl benzylhydrogenated tallow ammonium (MtMB), Mt modified with dimethyl benzylhydrogenated tallow ammonium containing JFE (MtMB + JFE), Mt modified with dimethyl dihydrogenated tallow ammonium (MtMD) and Mt modified with dimethyl dihydrogenated tallow ammonium containing JFE (MtMD + JFE).

### 2.3. Characterizations of Potential Active and pH-Sensitive Nanopackaging

#### 2.3.1. X-Ray Diffraction (XRD)

An X-Pert Pro diffractometer (Almelo, Netherlands) operating at 40 kV and 40 mÅ, with Cu K_α_ radiation (λ = 1.5406 Å), was used to obtain the XRD patterns of the Mts. Samples of finely ground Mt powder were placed in horizontal glass holders. XRD patterns were recorded at a scanning speed of 0.5°/min in an angular range of 2*θ* = 2° to 10°. The distances between the planes of the crystals *d* (Å) were then calculated from the diffraction angles (°) measured from the XRD patterns, following Bragg’s law [13]:(1)$$d=n*\lambda *{\left(2*\sin\theta \right)}^{-1}$$
where *n* is the order of reflection, and *λ* the wavelength of Cu K_α_ radiation. For the calculations, *n* = 1 was used. The differences between the interlayer distances (Δ_id_) of the samples tested were determined, considering as a reference the interlayer distance of the NMt (*d*_NMt_) as follows:(2)$$\Delta_{\mathrm{id}\;}=d_{c}-d_{\mathrm{NMt}}$$
where *d*_c_ is the interlayer distance of each Mt sample:

#### 2.3.2. Thermogravimetric Analysis (TGA)

A thermal analyzer (TA Instruments) Model TGA Q500 (Hüllhorst, Germany), at a heating rate of 10 °C/min from room temperature (approx. 30 °C) to 900 °C under nitrogen atmosphere, was used to carry out the TGA essays. The Mt mass varied between 10 and 28 mg. The mass fraction of JFE (*X*_JFE_) contained in the Mts was determined as follows:(3)$$X_{\mathrm{JFE}}=\frac{Rw_{n+\mathrm{JFE}}-Rw_{n}}{Rw_{\mathrm{JFE}}-Rw_{n}}$$
where *Rw_n+_*_JFE_ is the residual mass of the JFE-containing Mts, *Rw_n_* is the residual mass of the Mts without JFE, and *R_w_*_JFE_ is the residual mass of the JFE. Residual mass values were recorded at 900 °C where the decomposition of the JFE was stable (~11.77%). Analyses were conducted *per* triplicate to guarantee repeatability, and the data were reported as mean values ± standard deviation (SD).

#### 2.3.3. Field Emission Scanning Electron Microscopy (FESEM)

An FESEM Supra55, Zeiss (Oberkochen, Germany) operating at an acceleration voltage of 3 kV was used to acquire the FESEM micrographs of the Mts. ImageJ software was utilized to determine the average size of the Mt particles by randomly choosing at least 5 FESEM images for each nanosystem. All of the Mt samples were sputter coated with a thin layer of gold for 35 s, using an Ar^+^ ion beam at an energy level of 3 kV and sputter rate of 0.67 nm/min, to guarantee electrical conduction and to diminish surface charging during the essays. The sputter rate was determined by employing a Ni/Cr multilayer standard.

#### 2.3.4. Moisture Content (MC)

A Moisture Analyzer, Model MA150 (Goettingen, Germany) was used to determine the MC of each of the Mts. Samples (~0.5 g) were dried at 105 °C to a constant mass. Three measurements *per* nanosystem of Mt were obtained, and the values were reported as % average moisture ± SD.

#### 2.3.5. Attenuated Total Reflectance Fourier Transform Infrared (ATR/FTIR) Spectroscopy

The infrared spectra of the Mt samples were recorded on a Nicolet 8700 (Thermo Scientific Instrument Co., Madison, WI, USA) equipped with a diamond ATR probe at an incident angle of 45°, over the range of 4000–600 cm^−1^, from 32 co-added scans at 4 cm^−1^ resolution. Approximately 10 mg of each of the finely ground Mt samples were placed on the sample holder. Each sample was scanned three times, and showed good reproducibility.

#### 2.3.6. Raman Spectroscopy

An Invia Reflex confocal Raman microscope (Renishaw, Wotton-under-Edge, UK), with an argon laser operating at a power level of 3 mW, was set to obtain the Raman spectra of the Mts. The Raman spectra were obtained at 785 nm. The microscope that was used worked with a 50× objective lens to focus the beam onto the sample. The integration time was 0.5 s, and the number of accumulations were 200. Spectral resolution and repeatability were better than 1 cm^−1^ and 0.2 cm^−1^, respectively. The Raman spectrum for each sample was acquired as a number of evenly distributed points. No thermal effects were observed in the samples during the measurements.

#### 2.3.7. Confocal Laser Scanning Microscopy (CLSM)

An inverted microscope (Nikon, Eclipse T*i* series, Tokyo, Japan) operating at 515 nm was used to further examine the developed nanosystems. The samples were observed without any additional treatment.

#### 2.3.8. Color

The CIE-*L***a***b** color properties of the Mts were recorded from a Macbeth^®^ colorimeter (Color-Eye 2445 model, illuminant D65 and 10° observer, New Windsor, New York, USAcity, state, country). Color difference (Δ*E**) values among the modified Mts, with and without added JFE, and the control (NMt), were determined utilizing Equation (4) [23]:(4)$$\Delta E=\sqrt{\Delta a^{2}+\Delta b^{2}+\Delta L^{2}}$$
where Δ*L* = *L**_control_ − *L**_sample_, Δ*a* = *a**_control_ − *a**_sample_ and *b**_control_ − *b**_sample_.

The yellowness index (*YI*) of the Mts was determined according to ASTM D1925-70 [27], while the whiteness index (*WI*) was calculated as follows [28]:(5)$$WI=100-{\left(100-L\right)}^{2}+a^{2}+b^{2}$$

The chromaticity (*C**) and hue angle (°*h*) values were analyzed following García-Tejeda et al. [29]. Three readings were taken for each Mt nanosystem. The data were reported as mean values ± SD.

#### 2.3.9. Response to pH Changes

With the purpose of determining the response of the Mt nanosystems to different pH stimuli, samples of each nanosystem (0.1 g of Mt) were placed in 4 mL solutions of pH = 1, 7 and 13, made from NaOH (0.1M) and HCl. (0.1M). Color changes in the Mts were then evaluated from images that were acquired with an 8.1-megapixel Cyber-shot Sony camera model DSC-H3 (Tokyo, Japan).

#### 2.3.10. DPPH^•^ Antioxidant Activity

The total antioxidant activity of each Mt nanosystem was obtained utilizing the 2,2-diphenyl-2-picrylhydrazyl (DPPH^•^) radical methodology employed by Molyneux [30], with a UV-visible spectrometer UV-1601 PC (Shimadzu Corporation, Kyoto, Japan) at 517 nm. The antioxidant activity of the Mts analyzed was reported as the DPPH^•^ inhibition percentage, and determined following Equation (6):(6)$$\%Inhibition=\left(\;A_{0}-A_{60}\right)*\;{\left(A_{0}\right)}^{-1}*100$$
where *A*_0_ and *A*_60_ are the absorbance values of the blank sample and the sample-containing radical, respectively. Three determinations were made *per* nanosystem. The results were expressed as mean values ± SD.

#### 2.3.11. Antimicrobial Activity

The antimicrobial activity of the Mts was tested by exactly following the agar diffusion methodology described by Gutiérrez et al. [23]. The zone of inhibition on solid media assay was utilized to determine the antimicrobial properties of the Mts against two typical pathogen microorganisms: *Escherichia coli* O157:H7 (32158, American Type Culture Collection—Gram-negative bacteria) and *Listeria monocytogenes innocua* (Gram-positive bacteria) provided by CERELA (Centro de Referencia de Lactobacilos, Tucumán, Argentina). Each assay was performed in triplicate on two separate experimental runs.

#### 2.3.12. Cytotoxicity Assay

Cell viability was evaluated using the (3-(4,5-dimethylthiazol-2-yl)-2,5-diphenyl tetrazolium bromide (MTT) method, which is based on the ability of viable cells to metabolically reduce a yellow tetrazolium salt (MTT; Invitrogen) to purple crystals of formazan [31]. This reaction is given when the mitochondrial reductases are active. The cell line used for this study was normal human lung fibroblasts (MRC-5, ATCC CCL-171). The cells were grown in 96-well plates (3 × 10^4^ cells/well) for 24 h at 37 °C. Approximately 10 mg of each Mt were weighed in Eppendorf tubes, and 1 mL of distilled water was then added to each tube, in order to obtain suspensions at 1% *w/v*. These suspensions were stored in the dark for 24 h at 5 °C, and then micro-filtered using a syringe and membrane filter. Solutions of each Mt nanosystem at two different concentrations (0.25% and 0.5%) were then prepared. For the mitochondrial viability bioassays, cell monolayers were incubated in one of the Mt solutions for 24 h at 37 °C. Thereafter, the monolayers were laved with 1 mL of fetal bovine serum (FBS, Internegocios S.A., Buenos Aires, Argentina) and fixed with methanol (99.8%, Biopak—product code: 1655.08, El Salvador, Argentina) at ambient temperature during 10 min. The cells were then dyed with a solution of methylene blue (Merck, Darmstadt, Germany) for 10 min. The dye solution was then discarded, and the plate was laved with water. After this, the absorbance was determined at 570 cm^−1^, using a Thermo Scientific model 2200 (Waltham, MA, USA) microplate reader. According to Leon et al. [32], this colorimetric bioassay is strongly correlated to cell proliferation measured by cell counting in a Neubauer chamber under the conditions described above. Results were reported as percent cell viability relative to natural Mt (control nanosystem, NMt). Three trials were performed *per* Mt nanosystem, and the average values ± SD were reported.

#### 2.3.13. Cell Morphology

In order to analyze any changes to cell morphology after the cytotoxicity assay, the samples were treated as outlined in Section 2.3.12, and then observed under an optical microscope (Olympus IX51, Tokyo, Japan) at 50×. At least three microphotographs of each sample were taken with an 8.0-megapixel video camera imaging system (Olympus IMAGE RS, Tokyo, Japan) attached to a personal computer.

### 2.4. Statistic Analysis

All of the resulting data were analyzed using OriginPro 8 (Version 8.5, Northampton, MA, USA) software. The data were first evaluated with an analysis of variance (ANOVA), and significant results were subsequently analyzed using Duncan’s multiple range tests (*p* ≤ 0.05) to compare mean values.

## 3. Results and Discussion

### 3.1. X-Ray Diffraction (XRD)

The NMt displayed a diffraction peak at 2*θ* = 7.10° associated with a 001 basal spacing of 12.5 Å (Figure 1). Similar results were reported by Tunç and Duman [33,34] for pure Mt (12.7 Å). The interlayer spacings of the MtMB and MtMD (*d*-values ≅ 18.5 Å and 26.5 Å, respectively) were notably wider compared to that of the NMt, representing an increase of about 6.0 Å and 14.0 Å, respectively (Table 1). This suggests that the dimethyl dihydrogenated tallow ammonium organo-modifying agent increased interlayer spacing to a greater degree than the dimethyl benzylhydrogenated tallow ammonium organo-modifying agent, possibly because the former is a smaller compound than the latter, and can thus penetrate the interlayer spacing of the Mt more easily. de Azeredo [35] indicated that this behavior is related to cation exchange reactions between inorganic cations and organic cations in Mts.

The 001 basal reflections from MtMB and MtMD increased from the addition of Jamaica flower extract (JFE—100% polymeric anthocyanin) (MtMB and MtMD nanosystems compared to their respective analogous nanosystems containing JFE (MtMB + JFE and MtMD + JFE)) (Figure 1). This demonstrates that polymeric anthocyanins were intercalated (nanopacked/nanoencapsulated/loaded) in the silicate interlayer spaces [36]. It is worth noting that this behavior was not observed for the NMt nanosystem (NMt nanosystem compared to their respective analogous nanosystem containing JFE (NMt + JFE). The reason for this may be because the polymeric anthocyanins were larger than the basal spacing of the NMt (12.5 Å), thus preventing the JFE from entering and increasing the interlayer spacing.

### 3.2. Thermogravimetric Analysis (TGA)

The TGA curves (Figure 2) were analyzed in order to detect variations in the thermal stability of the nanosystems assessed; they were also used to determine the mole fraction of the JFE (*X*_JFE_) that was loaded into the Mts. Three stages of thermal degradation in the Mt nanosystems were observed: the first, from ambient temperature (30 °C) up to around 150 °C, was attributed to the evaporation of physisorbed water molecules; the second, between 150 °C and 450 °C, related to the slow loss of water molecules that were initially occluded between the interlayers; and the third, from 650 °C, resulted from the dehydroxylation of the structural OH groups of the Mts [23].

In the temperature region between 30 °C and 120 °C, the TGA curves of the JFE-containing Mt nanosystems were slightly displaced towards lower temperatures with regard to those of the Mt nanosystems without JFE. This suggests that possibly the polar sites (Lewis sites) available on the Mts were sterically hindered by polymeric anthocyanins, thereby weakening and reducing Mt–water molecule interactions [37].

The NMt nanosystem exhibited thermal behavior similar to that reported by Merino et al. [38] for natural bentonite, and had a lower thermal resistance than the organo-modified Mt nanosystems. This is consistent with Zhang et al. [39], who indicated that organoammonium groups grafted on the silicates show high thermal stability, and only begin to degrade at 400 °C. This phenomenon was more pronounced in the organo-Mt nanosystems that were modified with dimethyl dihydrogenated tallow ammonium, compared to the nanosystems that were modified with dimethyl benzylhydrogenated tallow ammonium. This suggests that the dimethyl-dihydrogenated-tallow-ammonium-modified reagent can enter the interlayer spacing more easily, and interact more strongly with the Mt than the dimethyl benzylhydrogenated tallow ammonium, thereby increasing the thermal resistance of the former. The JFE-containing Mt nanosystems (NMt + JFE, MtMB + JFE and MtMD + JFE) showed a lower heat resistance compared to their analogous nanosystems without added JFE (NMt, MtMB and MtMD). This fact is possibly related to the thermal degradation of the interleaved organic matter that was contained in the interlayer spaces of the Mts.

Interestingly, during the last stage of thermal degradation, MtMD showed a greater mass loss than MtMB, suggesting that there was more organic material packed within the interlayer spaces of the Mt organo-modified with dimethyl dihydrogenated tallow ammonium. This is in line with the results discussed so far.

The maximum thermal degradation temperature of the JFE (100% polymeric anthocyanin) was around 211 °C. According to the calculations made from the TGA curves, NMt + JFE did not nanopack the polymeric anthocyanins found in the JFE (Table 1). This agrees with the results obtained from the XRD diffractograms (see Section 3.1). The highest mole fraction of JFE (*X*_JFE_) nanopacked into the Mts was obtained for MtMB + JFE (∼0.10) (Table 1), whereas MtMD + JFE only nanopacked half of this amount (*X*_JFE_∼0.05). These results can be explained as follows: the organo-modification of Mt with dimethyl benzylhydrogenated tallow ammonium (MtMB) caused enough of an increase in the interlayer spacing to enable the entry of the polymeric anthocyanins. In addition, few modifying agent molecules were found within the interlayer spacing, thus leaving room for more JFE molecules to enter into it. In contrast, in MtMD + JFE, the molecules of the modifying agent (dimethyl dihydrogenated tallow ammonium; MtMD) easily entered the Mt structure. Therefore, the molar volume of polymeric anthocyanins nanopacked in MtMD + JFE was lower, despite having greater interlayer spacing.

### 3.3. Field Emission Scanning Electron Microscopy (FESEM)

The NMt nanoparticles were transformed slightly, from a spherical morphology (NMt, Figure 3a) to mainly irregular morphologies, either after organo-modification or when the JFE was added. The average size of the Mt particles also increased, from around 12 ± 2 μm (NMt and NMt + JFE) to 30 ± 6 μm, for the other nanosystems evaluated. Notwithstanding, no statistically significant variations among the particle sizes of the organo-modified Mts, with or without the addition of JFE, were observed, i.e., both organo-modifications led to an increase in particle size, regardless of the modifying agent used. These results fit well with the increase in interlayer spacing for these nanosystems, which was evidenced from the XRD patterns. Öztop and Shahwan [40] also reported that alkaline hydrothermal treatment-modified Mt has a similar behavior to that described above.

### 3.4. Moisture Content (MC)

The MC significantly differed among the treatments (*p* ≤ 0.05) as follows: MtMD < MtMB < NMt (Table 1). This shows that both organo-modifying agents reduced the hydrophilicity of the Mt nanosystems, with this being more significant when dimethyl dihydrogenated tallow ammonium was used compared to dimethyl benzylhydrogenated tallow ammonium. Since the former (MtMD) is less polar than the latter (MtMB), it seems reasonable to suppose that it compensates polar sites (Lewis sites) within the Mt structure, thus decreasing the Mt nanosystems’ hydrophilicity. This is in line with the TGA results (see Section 3.2). The MC of the JFE-containing Mts was statistically higher (*p* ≤ 0.05) than their respective analogous Mt nanosystems without JFE. This is in line with results published by Gutiérrez et al. [23] for natural and modified nano-clays, with and without added blueberry extract (BE), and suggests that the addition of JFE increases the vulnerability of Mt nanosystems to water uptake from the ambient environment.

### 3.5. Attenuated Total Reflectance Fourier Transform Infrared (ATR/FTIR) Spectroscopy

The FTIR spectra for the manufactured Mt nanosystems over the entire absorption range (Figure 4A) showed absorption peaks at around 3624 cm^−1^, which can be related to the structural OH groups in the Mts, and the bands centered at 3422 cm^−1^ associated with the stretching and bending vibrations of the hydroxyl (O-H) groups of the free water molecules physisorbed in the Mt nanosystems [38,41]. The peak intensity at 3422 cm^−1^ was greater in the JFE-containing Mt nanosystems than their analogous nanosystems without JFE (Figure 4B), suggesting that available cations were exchanged for protons from polar groups during JFE incorporation, thus raising the number of available OH groups [41]. This fits well with the MC results (see Section 3.4) that showed that the JFE-containing Mt nanosystems were more hydrophilic, i.e., higher MC values correlated with stronger absorption bands that were associated with free water and O-H groups. The bands situated at 2917 cm^−1^ and 2844 cm^−1^ corresponded to CH_2_ asymmetric and CH_2_ symmetric stretching vibrations within the modifying agents, and another band centered at 1468 cm^−1^ was associated with the deformation vibrations of CH_2_/CH_3_ that were also within the modifying agents [38]. The absorption bands positioned at 990 cm^−1^ were associated to Si-O groups in plane vibration, while the absorption bands at 916, 880 and 800 cm^−1^ were assigned to Al-Al-OH, Al-Fe-OH and Al-Mg-OH bending vibrations, respectively [42,43,44]. A decrease in the intensities of the bands located at 814 cm^−1^ (O-Si-O asymmetric stretching) and 990 cm^−1^ (Si-O stretching) was also evidenced, and was attributed to variations in the Si environment.

The FTIR spectrum for the JFE (100% polymeric anthocyanin) showed all of the absorption peaks that corresponded to the active compound, all of which have been well characterized elsewhere in the literature [45].

### 3.6. Raman Spectroscopy and Confocal Laser Scanning Microscopy (CLSM)

In order to learn more about the structure of the manufactured Mts, Raman spectra were acquired (Figure 5). However, neither the organo-modified Mt nanosystems nor those with added JFE showed clear bands; thus, we were unable to discover more details about the structure of the Mt nanosystems. Gutiérrez et al. [23] indicated that the fluorescence phenomenon limits the acquisition of Raman spectra. To confirm this hypothesis, the Mt nanosystems were scanned using confocal laser scanning microscopy (CLSM) (Figure 6). The CLSM images confirmed the self-fluorescence of the Mt nanosystems developed. This is possibly related to multiple oxygen-containing functional groups. These functional groups can induce numerous localized energy levels within the n-π* gap, which decentralizes the excited electrons [46]. Notably, the fluorescence in the JFE-containing Mt nanosystems (NMt + JFE, MtMB + JFE and MtMD + JFE) was higher compared to their analogous nanosystems without added JFE (NMt, MtMB and MtMD). These fluorescence-inducing properties could make JFE (100% polymeric anthocyanin) promising for the development of theragnostic nanosystems, since, as will be discussed in the following sections, it could be used in the treatment of different diseases, and at the same time be localized by the fluorescence phenomenon. On a different note, and as expected, the morphology of the Mt nanosystems observed by CLSM (see Figure 6) was similar to that observed using FESEM (see Figure 3).

### 3.7. Color

The results of the color analyses (Table 1) revealed that both organo-modification and/or the addition of JFE resulted in darker Mt nanosystems (see *L** values). The *a** values showed that all of the Mt nanosystems tended towards a red coloration; however, no clear tendency due to organo-modification or the incorporation of JFE was observed. The *b** values, notwithstanding, showed that although all of the Mt nanosystems tended towards yellow, this was notoriously less pronounced (*p* ≤ 0.05) in those containing JFE, i.e., the addition of JFE produced less yellow nanosystems.

According to Obón et al. [47], obvious color differences identifiable by the human eye yield color difference (Δ*E*) values higher than 5. In this study, all of the treatments carried out on the NMt led to changes in coloration that were easily detectable by the naked eye.

The NMt had the highest whiteness index (*WI*) compared to the treated Mt nanosystems, which is consistent with the *L** values.

A decline in the chromaticity (*C**) values was appreciated by incorporating JFE into the Mt nanosystems. Similar results were obtained by Gutiérrez et al. [23] for nano-clays that contained polymeric anthocyanins extracted from blueberries. These *C** values were not drastically altered by the organo-modifications carried out on the Mt nanosystems.

The hue angles (°*h*) of the clay nanosystems were correctly located in the quadrants of the CIE *L***a***b** color chart.

### 3.8. Response to pH Changes

The images of the color variations in the Mt nanosystems under different pH conditions (Figure 7) revealed that the JFE-containing Mts (NMt + JFE, MtMB + JFE and MtMD + JFE) showed a bathochromic effect, i.e., they changed color depending on the pH. These Mt nanosystems had a pink coloration at pH = 1, which was provoked by the formation of the flavylium cation (also called 2-phenyl-benzopyrylium, and consists of two aromatic groups: a benzopyrilium and a phenolic ring) or the oxonium ion, which are the most stable polymeric anthocyanin structures. At pH = 13, however, the JFE-containing Mt nanosystems showed a yellow coloration, possibly because of the quinoidal structure of the polymeric anthocyanins that formed at pH > 8 [48]. Our research group previously developed pH-sensitive nano-clays by incorporating polymeric anthocyanins extracted from blueberries. However, the color changes observed were different from those of the current study [23]. Color changes due to pH were produced by hydroxyl groups of the phenolic rings, and by the benzopirilium [49,50]. This suggests that the flavonoid structures of polymeric anthocyanins differ, depending on the plant source. It is worth noting that at least 20 different polymeric anthocyanin structures are known, and it is estimated that there may be as many as 150 [51]. The bathochromic or pH-sensitive effect observed in the JFE-containing Mt nanosystems is very interesting for the development of composite materials aimed at manufacturing pH-sensitive (“intelligent”) food nanopackaging, or as nanosensors for theragnostic devices.

It should also be noted that the JFE at pH = 13 produced a slight emission of gases. This is because at alkaline pH, the loss of a proton from, and the addition of water to, the polymeric anthocyanin structure leads to a balance between the pseudobase hemicetal (carbinol) and chalcone (an open chain). Both hemicetal and chalcone are quite unstable forms and, at pH values higher than 7, are rapidly degraded by oxidation with air [49,52]. This means that at pH = 13, these JFE-containing Mt nanosystems undergo an irreversible color change.

Fascinatingly, not only did the color of the JFE-containing Mt nanosystems change, that of the solution they were in also changed, slightly. This verifies the diffusion of polymeric anthocyanin molecules towards the aqueous medium. Hence, polymeric anthocyanins can work as tracers of events taking place within the Mt nanosystems. This suggests that the polymeric anthocyanins that were nanopackaged within the interlayer spaces created van der Waals-type interactions between the Mts and the pigment. Otherwise, the chromophore groups of the polymeric anthocyanins would prevent pH-induced color variations in the Mt nanosystems.

### 3.9. DPPH^•^ Antioxidant Activity and Antimicrobial Activity

The results of the antioxidant activity study showed that JFE is a highly active substance (∼77% inhibition of the DPPH^•^ radical—Figure 8). Similar results were reported by other authors elsewhere [3]. As expected, the Mt nanosystems without added JFE showed negligent oxidant activity (∼1.0%). Gutiérrez et al. [23] reported similar values for this type of clay system. In contrast, all JFE-containing Mt nanosystems showed an antioxidant activity of around 50%. This confirms the stabilizing effect of the Mt nanosystems on the polymeric anthocyanins found in the JFE. Similar behavior was observed by Ribeiro et al. [20] for anthocyanins that were obtained from acerola juice, and stabilized with Mt. Importantly, all of the polymeric anthocyanin-stabilizing Mt nanosystems displayed similar antioxidant activities, suggesting that the polymeric anthocyanins were stabilized whether or not they intercalated within the interlayer spacing of the Mts (see Section 3.1). This meant that they could also be stabilized by chemical interactions that occurred between them and the surfaces of the Mt nanosystems.

The antioxidant activity of these polymeric anthocyanin-stabilizing Mt nanosystems represents a promising alternative for the treatment of different diseases, as has been discussed in the literature. This means that JFE could also be utilized for the production of medicinal and functional foods [1].

In regards to the antimicrobial properties of the Mt nanosystems, other authors have demonstrated that both JFE and two commercial modified Mt nanosystems, Cloisite 20A and Cloisite 30B, show antimicrobial activity [35,53]. In this study, however, the polymeric anthocyanin-stabilizing Mt nanosystems that were developed had no antimicrobial activity against either *Escherichia coli* O157:H7 or *Listeria monocytogenes innocua*. According to Gutiérrez et al. [23], nano-clays that contained BE within their structure exhibited a similar trend. In spite of this, no microbiological growth was recorded for the polymeric anthocyanin-stabilizing Mt nanosystems.

### 3.10. Cytotoxicity Assay and Cell Morphology

The cell viability assays showed that natural Mts (NMt and NMt + JFE) are not cytotoxic to human cells, even at doses as high as 0.5% *w/v* (Figure 9). These results were validated through morphological observations of cells that were exposed to these two Mt nanosystems: no damage to cell ultrastructure was observed (Figure 10a,b). In fact, cell viability was significantly greater (*p* ≤ 0.05) after exposure to the higher dose (0.5% *w/v*) of the polymeric anthocyanin-stabilizing Mt nanosystem (NMt + JFE), than it was after exposure to the lower dose (0.25% *w/v*). This suggests that a dose of JFE at 0.5% *w/v* could be beneficial for the growth of healthy human cells. In contrast, the Mt nanosystems modified with dimethyl benzylhydrogenated tallow ammonium (MtMB and MtMB + JFE) showed high cytotoxicity, even at the lower dose (0.25% *w/v*), i.e., cell viability was reduced after exposure to these systems. These results were confirmed by optical microscopy (Figure 10c,d), which revealed that the Mt nanosystems organo-modified with dimethyl benzylhydrogenated tallow ammonium destroyed the cell membrane, leaving the nucleus exposed, and leading to a collapse of the cell structure. These results are regrettable, since the MtMB + JFE polymeric anthocyanin-stabilizing nanosystem nanopacked the highest amounts of JFE within its structure (see Section 3.2). Similar behavior was reported in the literature for clays that were organo-modified by silylation [24]. In contrast, the Mt systems organo-modified with dimethyl dihydrogenated tallow ammonium (MtMD and MtMD + JFE) had a similar effect on cell viability as the NMt nanosystems (NMt and NMt + JFE), except that a slight decrease in viability was evidenced at the lower dose (0.25% *w/v*). This was also observed in samples examined under the optical microscope (Figure 10e,f).

## 4. Conclusions

Montmorillonites (Mt) were shown to have the capacity to form polymeric anthocyanin-stabilizing nanosystems by two mechanisms: (1) intercalation of the bioactive compound within the interlayer spacing of the Mts, and (2) van der Waals-type chemical interactions between the polymeric anthocyanins and the outer surfaces of the Mts. Unfortunately, the Mts that were organo-modified with dimethyl benzylhydrogenated tallow ammonium (MtMB and MtMB + JFE) were cytotoxic for the cell line evaluated (normal human lung fibroblast, MRC-5, ATCC CCL-171), thus limiting their application as food additives. The other two polymeric anthocyanin-stabilizing nanopackaging systems that were developed, however, show promise for the development of food additives and nano-sensors, with applications in the food and biotechnology industries. This is due to their pH-sensitive behavior, self-fluorescence properties and antioxidant activity, although sadly, they did not show antimicrobial activity. Remarkably, the polymeric anthocyanin-stabilizing and nanopackaging systems were more hydrophilic compared to their analogous nanosystems without polymeric anthocyanins. Finally, the novelty of this study highlights the importance and the crucial point of the toxicological assessment of nanopackaging materials; as demonstrated here, extraordinary food packaging nanosystems able to stabilize and encapsulate active compounds can be obtained, but their food toxicity may limit their use as food contact material. To the best of the authors’ knowledge, these results provide a significant advance in the understanding of this field, since very few studies have characterized and evaluated the toxicity of these kinds of materials.

## Figures and Tables

**Figure 1 polymers-14-04881-f001:**
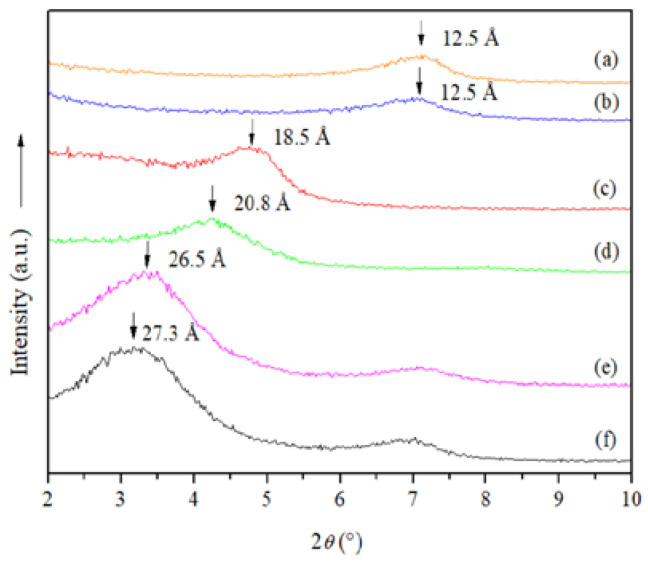
X-ray diffraction patterns of the different Mt nanosystems studied: (a) natural montmorillonite (NMt), (b) natural montmorillonite-containing Jamaica flower extract (NMt + JFE), (c) montmorillonite modified with dimethyl benzylhydrogenated tallow ammonium (MtMB), (d) montmorillonite modified with dimethyl benzylhydrogenated tallow ammonium containing Jamaica flower extract (MtMB + JFE), (e) montmorillonite modified with dimethyl dihydrogenated tallow ammonium (MtMD) and (f) montmorillonite modified with dimethyl dihydrogenated tallow ammonium containing Jamaica flower extract (MtMD + JFE).

**Figure 2 polymers-14-04881-f002:**
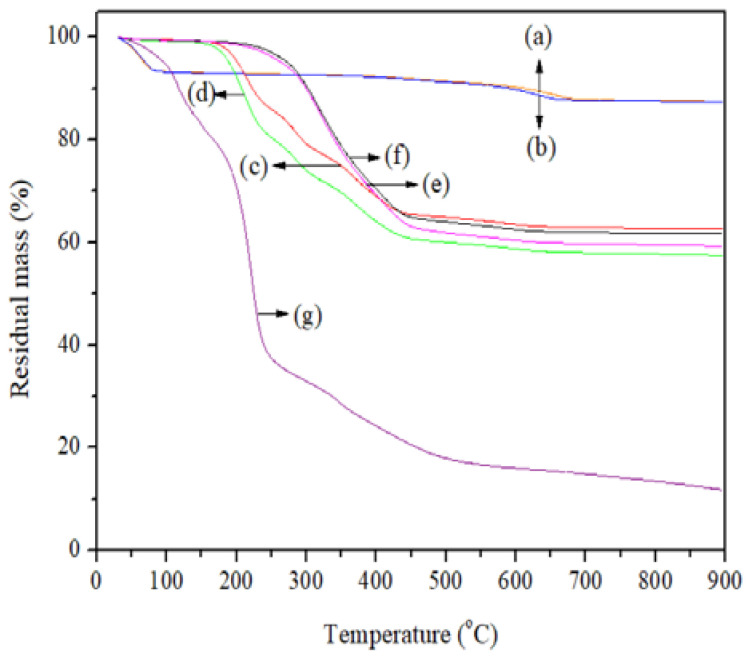
TGA curves of the Mt nanosystems and the extract studied: (a) natural montmorillonite (NMt), (b) natural montmorillonite containing Jamaica flower extract (NMt + JFE), (c) montmorillonite modified with dimethyl benzylhydrogenated tallow ammonium (MtMB), (d) montmorillonite modified with dimethyl benzylhydrogenated tallow ammonium containing Jamaica flower extract (MtMB + JFE), (e) montmorillonite modified with dimethyl dihydrogenated tallow ammonium (MtMD), (f) montmorillonite modified with dimethyl dihydrogenated tallow ammonium containing Jamaica flower extract (MtMD + JFE) and (g) Jamaica flower extract (JFE).

**Figure 3 polymers-14-04881-f003:**
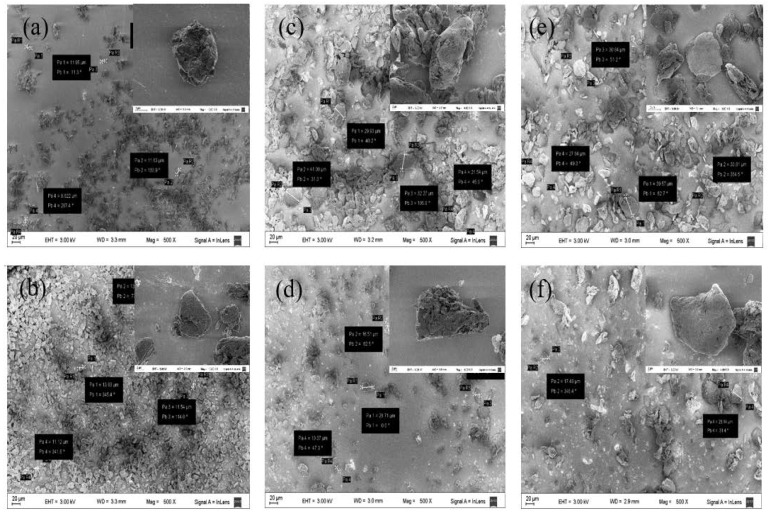
FESEM micrographs of the different Mt nanosystems studied: (**a**) natural montmorillonite (NMt), (**b**) natural montmorillonite containing Jamaica flower extract (NMt + JFE), (**c**) montmorillonite modified with dimethyl benzylhydrogenated tallow ammonium (MtMB), (**d**) montmorillonite modified with dimethyl benzylhydrogenated tallow ammonium containing Jamaica flower extract (MtMB + JFE), (**e**) montmorillonite modified with dimethyl dihydrogenated tallow ammonium (MtMD) and (**f**) montmorillonite modified with dimethyl dihydrogenated tallow ammonium containing Jamaica flower extract (MtMD + JFE). Some particle sizes obtained can be seen in the boxes from each image.

**Figure 4 polymers-14-04881-f004:**
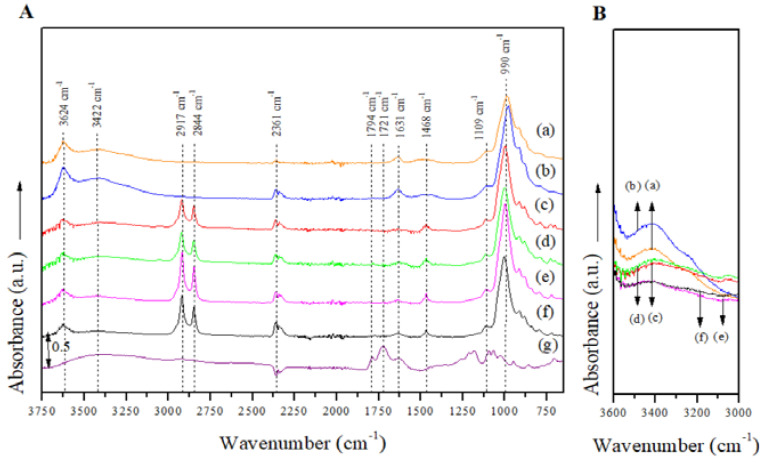
(**A**) FTIR spectra of the Mt nanosystems and the extract studied in the entire absorption range: (a) natural montmorillonite (NMt), (b) montmorillonite modified with dimethyl benzylhydrogenated tallow ammonium (MtMB), (c) montmorillonite modified with dimethyl dihydrogenated tallow ammonium (MtMD), (d) natural montmorillonite containing Jamaica flower extract (NMt + JFE), (e) montmorillonite modified with dimethyl benzylhydrogenated tallow ammonium containing Jamaica flower extract (MtMB + JFE), (f) montmorillonite modified with dimethyl dihydrogenated tallow ammonium containing Jamaica flower extract (MtMD + JFE) and (g) Jamaica flower extract (JFE). (**B**) FTIR spectra in the absorption range corresponding to C-O groups (OH stretching) of the Mt nanosystems and the extract evaluated: (a) NMt, (b) MtMB, (c) MtMD, (d) NMt + JFE, (e) MtMB + JFE and (f) MtMD + JFE.

**Figure 5 polymers-14-04881-f005:**
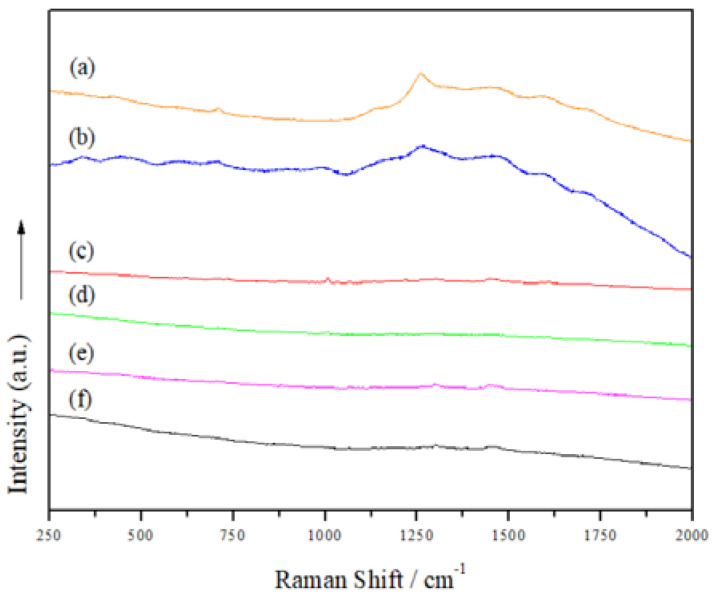
Raman spectra of the different Mt nanosystems studied: (a) natural montmorillonite (NMt), (b) natural montmorillonite containing Jamaica flower extract (NMt + JFE), (c) montmorillonite modified with dimethyl benzylhydrogenated tallow ammonium (MtMB), (d) montmorillonite modified with dimethyl benzylhydrogenated tallow ammonium containing Jamaica flower extract (MtMB + JFE), (e) montmorillonite modified with dimethyl dihydrogenated tallow ammonium (MtMD) and (f) montmorillonite modified with dimethyl dihydrogenated tallow ammonium containing Jamaica flower extract (MtMD + JFE).

**Figure 6 polymers-14-04881-f006:**
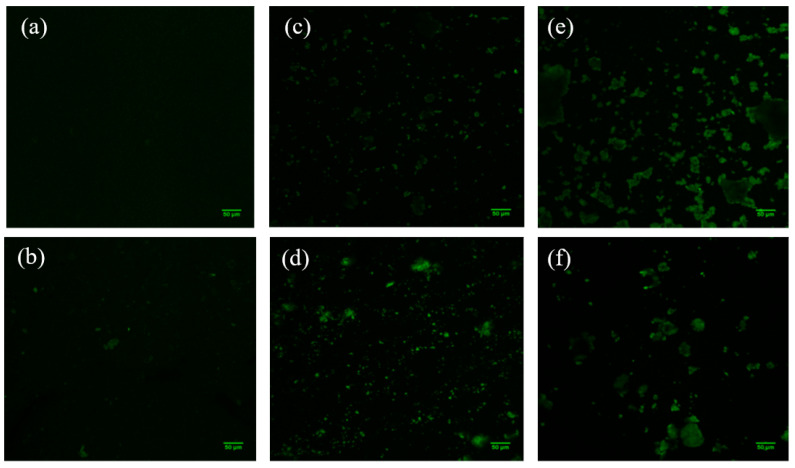
CLSM micrographs of the different Mt nanosystems studied: (**a**) natural montmorillonite (NMt), (**b**) natural montmorillonite containing Jamaica flower extract (NMt + JFE), (**c**) montmorillonite modified with dimethyl benzylhydrogenated tallow ammonium (MtMB), (**d**) montmorillonite modified with dimethyl benzylhydrogenated tallow ammonium containing Jamaica flower extract (MtMB + JFE), (**e**) montmorillonite modified with dimethyl dihydrogenated tallow ammonium (MtMD) and (**f**) montmorillonite modified with dimethyl dihydrogenated tallow ammonium containing Jamaica flower extract (MtMD + JFE).

**Figure 7 polymers-14-04881-f007:**
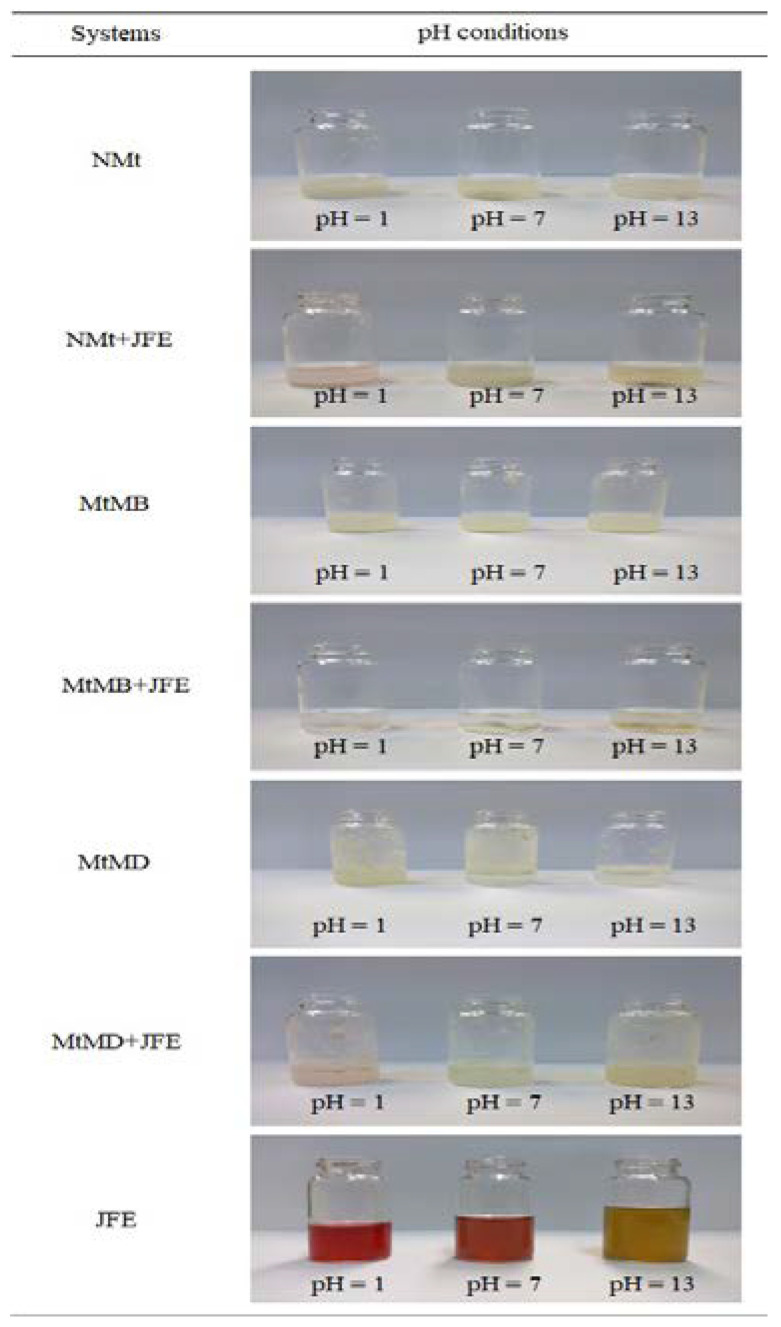
Response of the Mt nanosystems evaluated under different pH conditions: natural montmorillonite (NMt), natural montmorillonite containing Jamaica flower extract (NMt + JFE), montmorillonite modified with dimethyl benzylhydrogenated tallow ammonium (MtMB), montmorillonite modified with dimethyl benzylhydrogenated tallow ammonium containing Jamaica flower extract (MtMB + JFE), montmorillonite modified with dimethyl dihydrogenated tallow ammonium (MtMD), montmorillonite modified with dimethyl dihydrogenated tallow ammonium containing Jamaica flower extract (MtMD + JFE) and Jamaica flower extract (JFE).

**Figure 8 polymers-14-04881-f008:**
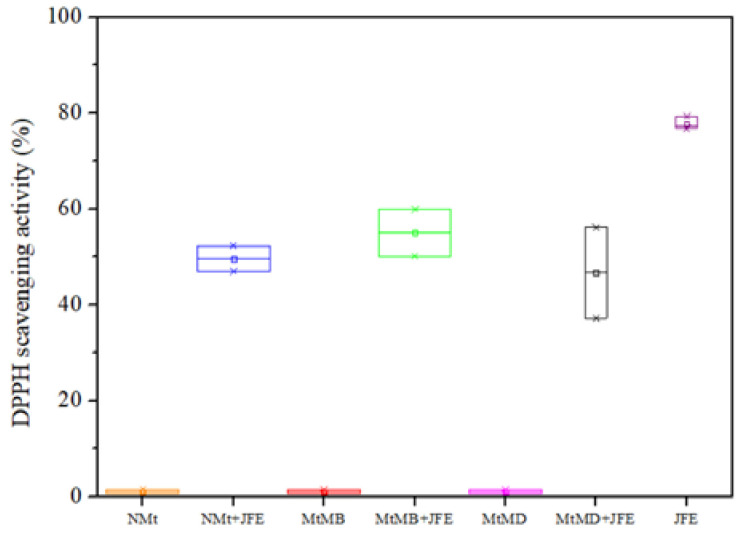
DPPH^•^ cavenging activity of the Mt nanosystems and the extract evaluated: natural montmorillonite (NMt), natural montmorillonite containing Jamaica flower extract (NMt + JFE), montmorillonite modified with dimethyl benzylhydrogenated tallow ammonium (MtMB), montmorillonite modified with dimethyl benzylhydrogenated tallow ammonium containing Jamaica flower extract (MtMB + JFE), montmorillonite modified with dimethyl dihydrogenated tallow ammonium (MtMD), montmorillonite modified with dimethyl dihydrogenated tallow ammonium containing Jamaica flower extract (MtMD + JFE) and Jamaica flower extract (JFE).

**Figure 9 polymers-14-04881-f009:**
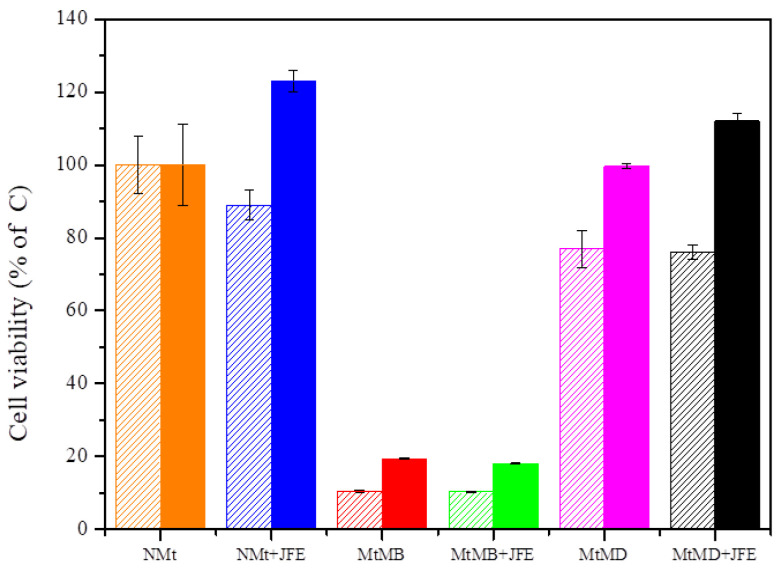
Cell viability of the different Mt nanosystems evaluated at two different doses (dose 0.25% *w/v*—segmented boxes; dose 0.5% *w/v*—solid boxes). Mt nanosystems: natural montmorillonite (NMt), natural montmorillonite containing Jamaica flower extract (NMt + JFE), montmorillonite modified with dimethyl benzylhydrogenated tallow ammonium (MtMB), montmorillonite modified with dimethyl benzylhydrogenated tallow ammonium containing Jamaica flower extract (MtMB + JFE), montmorillonite modified with dimethyl dihydrogenated tallow ammonium (MtMD) and montmorillonite modified with dimethyl dihydrogenated tallow ammonium containing Jamaica flower extract (MtMD + JFE).

**Figure 10 polymers-14-04881-f010:**
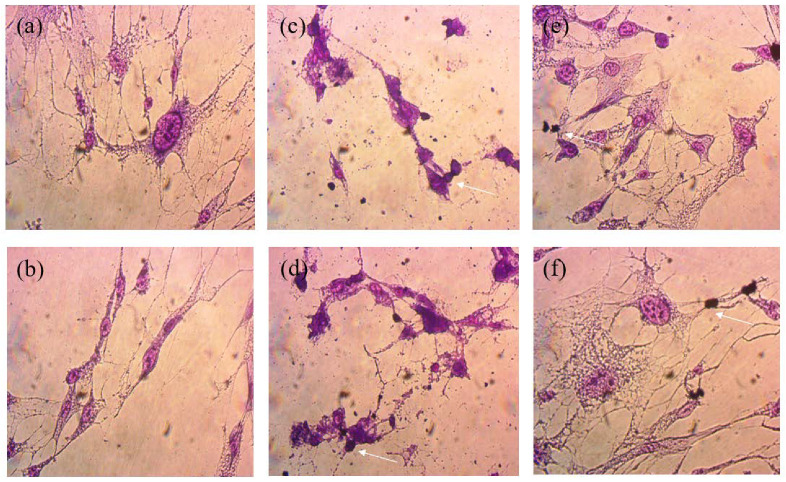
Optical micrographs (at 50× of magnification) of the morphology of cells exposed to the different Mt nanosystems studied: (**a**) natural montmorillonite (NMt), **(b**) natural montmorillonite containing Jamaica flower extract (NMt + JFE), (**c**) montmorillonite modified with dimethyl benzylhydrogenated tallow ammonium (MtMB), (**d**) montmorillonite modified with dimethyl benzylhydrogenated tallow ammonium containing Jamaica flower extract (MtMB + JFE), (**e**) montmorillonite modified with dimethyl dihydrogenated tallow ammonium (MtMD) and (**f**) montmorillonite modified with dimethyl dihydrogenated tallow ammonium containing Jamaica flower extract (MtMD + JFE). The arrows denote cell death of normal human lung fibroblast cells as an example. Many other similar dead cells can be seen from the same field of view in each image.

**Table 1 polymers-14-04881-t001:** Differences in interplanar distances (Δ_id_), mole fractions of the Jamaica flower extract (*X*_JFE_) loaded within the Mt nanosystems, moisture content (MC), and color parameters of the different Mt nanosystems.

Parameter	NMt	NMt + JFE	MtMB	MtMB + JFE	MtMD	MtMD + JFE
**Δ_id_ (Å)**	-	0.0 ± 0.1 ^a^	6.0 ± 0.1 ^b^	8.3 ± 0.1 ^c^	14.0 ± 0.1 ^d^	14.8 ± 0.1 ^e^
** *X* ** ** _JFE_ **	-	0.00 ± 0.01 ^a^	-	0.10 ± 0.01 ^c^	-	0.05 ± 0.01 ^b^
**MC (%)**	5.6 ± 0.7 ^d^	9.5 ± 0.4 ^e^	1.8 ± 0.2 ^b^	2.6 ± 0.2 ^b,c^	1.4 ± 0.1 ^a^	2.3 ± 0.3 ^b^
** *L** **	83 ± 5 ^d^	49 ± 2 ^a^	65 ± 5 ^b^	69 ± 2 ^b,c^	60.4 ± 0.6 ^b^	68 ± 5 ^b,c^
** *a** **	1.9 ± 0.1 ^c^	1.48 ± 0.07 ^b^	1.20 ± 0.08 ^a^	1.19 ± 0.01 ^a^	1.43 ± 0.08 ^b^	1.99 ± 0.01 ^c^
** *b** **	10.6 ± 0.7 ^d^	6.3 ± 0.1 ^a^	11.7 ± 0.7 ^d^	8.2 ± 0.1 ^c^	12.0 ± 0.3 ^d,e^	7.70 ± 0.09 ^b^
**Color difference (** **Δ*E*)**	0.00 ± 0.00 ^a^	34 ± 2 ^d^	18 ± 5 ^b^	14 ± 2 ^b^	22.4 ± 0.6 ^b,c^	15 ± 5 ^b^
**Whiteness** **Index (*WI*)**	80 ± 4 ^d^	49 ± 2 ^a^	63 ± 5 ^b^	68 ± 2 ^b,c^	58.6 ± 0.7 ^b^	67 ± 4 ^b,c^
** *C** **	10.8 ± 0.7 ^d^	6.5 ± 0.1 ^a^	11.7 ± 0.7 ^d^	8.3 ± 0.1 ^c^	12.1 ± 0.4 ^d,e^	7.95 ± 0.09 ^b^
***h* (^o^)**	190.4 ± 0.2 ^d^	193.1 ± 0.4 ^e^	185.84 ± 0.01 ^a^	188.22 ± 0.01 ^c^	186.8 ± 0.2 ^b^	194.5 ± 0.2 ^f^

Equal letters in the same row indicate no statistically significant differences (*p* ≤ 0.05). Mt nanosystems studied: natural montmorillonite (NMt), natural montmorillonite-containing Jamaica flower extract (NMt + JFE), montmorillonite modified with dimethyl benzylhydrogenated tallow ammonium (MtMB), montmorillonite modified with dimethyl benzylhydrogenated tallow ammonium containing Jamaica flower extract (MtMB + JFE), montmorillonite modified with dimethyl dihydrogenated tallow ammonium (MtMD) and montmorillonite modified with dimethyl dihydrogenated tallow ammonium containing Jamaica flower extract (MtMD + JFE).

## Data Availability

Transparency data associated with this article can be found in the online version at https://doi.org/10.17632/hxjftyt7k8.1.

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
