# Peer review of "Active and pH-Sensitive Nanopackaging Based on Polymeric Anthocyanin/Natural or Organo-Modified Montmorillonite Blends: Characterization and Assessment of Cytotoxicity"

_polymers, 2022, doi:10.3390/polym14224881_

Round 1

Reviewer 1 Report

Dear Authors,

the reviewed work concerns the interesting problem of introducing anthocyanins between montmorillonite layers and thus obtaining active and pH-sensitive mixtures. I agree that such blends can be used to develop food additives and nanosensors as you have proposed. In my opinion the article was just well written, a lot of experiments were done and the results were well discussed.

However, there are a few elements that need to be clarified.

1. Was the cation exchange capacity (CEC) Mt (measured by the methylene blue method) measured by the Authors? If so, the methodology should be given (line 83).

2. What were the blending conditions of JFE and Mts (line 99)?

3. The methodology section does not describe how the Whiteness Index (WI) was calculated (Table 1). (line 251-252).

Reviewer 2 Report

In the present manuscript the authors reported the evaluation and characterization of different systems, namely an active and pH-sensitive nanopackaging based on polymeric 

anthocyanin/natural or organo-modified montmorillonite blends. For their characterization (structural, thermal, morphological, physicochemical, antioxidant and antimicrobial properties) the authors applied different techniques specifically, X-Ray Diffraction (XRD), Thermogravimetric Analysis (TGA), Field Emission Scanning Electron Microscopy (FESEM), Moisture Content (MC), Attenuated Total Reflectance Fourier Transform Infrared (ATR/FTIR) Spectroscopy, Raman Spectroscopy, Confocal Laser Scanning Microscopy (CLSM), Color, Response to pH Changes, DPPH Antioxidant Activity, Antimicrobial Activity. The authors also evaluate the toxicity.

The paper explores a very interesting field the active and intelligent packaging, which has very useful applications in the food industry.

The manuscript is well-organized, and the objectives were clearly defined.

However, some aspects need to be revised to improve the manuscript version. The recommendations are the following.

-       Introduction: include a summary of devices/intelligent packaging developed for similar applications and reported in the literature in the past few years would be advisable

-       The quality of figures should be improved.

-       Conclusions: to highlight the novelty and advantages of this intelligent packaging compared to those already described in the literature it would be convenient.

I recommend clarify these aspects before publication.

Reviewer 3 Report

This is an interesting study about active and pH-sensitive nanopackaging. I strongly recommend it for publication after the following minor points are addressed.

1. The authors should improve the resolution of all the figures. The text in the figures is not clearly seen.

2. Line 37-39, one recent study (Pharmaceutics 14.11 (2022): 2272) should be included to support such a claim.

3. What is the molecular weight of polymeric anthocyanins?

4. How is the stability of these materials?

Reviewer 4 Report

Gutierrez et al. present a manuscript of interest and novel results. The authors present strong conclusions supported by the results.

The authors adequately and precisely carry out a justification of the study without precise details. The techniques presented are adequate and allow a coherent discussion of the results.

Authors need to improve minor points.

-The authors must carry out an independent discussion. The authors present results with enough entity for the manuscript to have its own discussion.

-In this sense, the authors must improve the quality of some parts of the results. Figure 8 and 9.

-The authors should improve the figure legends with more data.

-In the images of cell cultures the magnifications are missing. I suggest authors use arrows to highlight findings.

-The authors must improve very extensively the use of English grammar.

-I suggest the authors include a graphic summary.
